# Keep Refining Your Discrete Diffusion Model: A Mixture of Absorbing and Uniform Processes

## Abstract

Discrete diffusion models (DDMs) present a promising alternative to the autoregressive models, and is advantageous in supporting bidirectional attention, parallel generation, and greater controllability. However, DDMs either use (i) an uniform diffusion process, which provides token-level refinement but may cause abrupt changes in meaning at the sequence level; or (ii) an absorbing diffusion process that ensures stable semantic evolution but sacrifices refinement at the token level after unmasking. To resolve this dilemma, we synergize the adva ntages of both with the Mixture of Absorbing and Uniform Diffusion (MAUD) model. MAUD constructs a novel state transition matrix to interpolate between the two diffusion processes, simultaneously achieving sequence-level semantic stability and gradual token-level refinement. Empirical results show that MAUD outperforms existing DDMs in both language generation and language understanding tasks.

## 1 Introduction

Autoregressive (AR) models have demonstrated revolutionary performance on many language tasks (Brown et al., 2020; Touvron et al., 2023). However, the sequential next-token generation paradigm lacks bidirectional modeling capabilities and precludes parallel token generation. Recently, inspired by the remarkable success of continuous diffusion models in various applications such as image generation (Ho et al., 2020; Song & Ermon, 2019; Rombach et al., 2022; Lipman et al., 2023; Esser et al., 2024) and audio synthesis (Liu et al., 2022; 2023), discrete diffusion models (DDMs) tailored for language tasks have also achieved notable progress (Austin et al., 2021; Lou et al., 2024; Sahoo et al., 2024; Shi et al., 2024; Schiff et al., 2024; Nie et al., 2025). Unlike continuous data, language is discrete, necessitating specialized approaches. DDMs are designed specifically for such data. By design, DDMs feature bidirectional attention, support parallel generation, and offer greater controllability. They demonstrate distinct advantages for structured sequence modeling (Sahoo et al., 2024; Schiff et al., 2024) and complex logical reasoning (Ye et al., 2025a).

Current DDMs can be broadly divided into two categories based on the underlying diffusion process: uniform diffusion models (UDMs; Schiff et al. (2024)) and absorbing diffusion models (ADMs; Sahoo et al. (2024); Shi et al. (2024)). UDMs corrupt data by transitioning any token to another in the vocabulary. This allows for gradual, token-level refinement but can cause abrupt changes in meaning at the sequence level. In contrast, ADMs introduce an absorbing state (known as the [MASK]), such that each token either remains unchanged or is replaced by the [MASK] token with a certain probability. This ensures stable semantic evolution at the sequence level, but sacrifices the step-by-step refinement at the token level. In particular, once a token is unmasked, it cannot be revised in subsequent steps.

Motivated by this observation, we propose the **Mixture of Absorbing and Uniform Diffusion** (MAUD) model. By interpolating between the absorbing and uniform diffusion processes, MAUD maintains semantic stability at the sequence level while preserving gradual refinement at the token level. Specifically, we construct a novel state transition matrix by interpolating between the absorbing and uniform diffusion processes. We then derive its forward noising process and reverse denoising process, together with its NELBO objective. Empirically, MAUD achieves state-of-the-art performance among DDMs on a diverse set of language generation and understanding tasks.

The main contributions of this paper is summarized as follows:

- We identify the limitations of purely absorbing or uniform diffusion processes.
- We introduce a novel discrete diffusion model that interpolates between absorbing and uniform processes, together with a formal derivation of its forward noising process, reverse denoising process, and training objective.
- Extensive experiments on various tasks and benchmark datasets demonstrate that the proposed method surpasses state-of-the-art discrete diffusion models.

**Notations.** Given a vocabulary of $N$ unique tokens, let $\mathcal{V} = \{\mathbf{z} \in \{0,1\}^N : \sum_i z_i = 1\} \subset \Delta^N$ be the space of one-hot tokens, and $\Delta^N$ is the $N$-dimensional simplex. We assume that the $N$-th token (category) corresponds to a special [MASK] token, and the corresponding one-hot vector is denoted $\mathbf{m} \in \mathcal{V}$. Additionally, let $\mathbf{I}$ be the identity matrix, $\mathbf{1} = \{1\}^N$ be the $N$-dimensional vector of all ones, and $\odot$ be the Hadamard product between two vectors. We define $\mathbf{z}^{1:L}$ as a sequence of $L$ tokens, where $\mathbf{z}^\ell \in \mathcal{V}$ for $\ell \in \{1, \ldots, L\}$. Let the set of all such sequences be $\mathcal{V}^L$. Finally, let $\mathrm{Cat}(\cdot; p)$ be the categorical distribution with probability vector $p \in \Delta^N$.

## 2 BACKGROUND: DISCRETE DIFFUSION MODELS

Let $T$ be the number of discrete time steps. We define a pair of adjacent normalized timestep functions $s(i) = \frac{i-1}{T}$ and $t(i) = \frac{i}{T}$ (with $s(i)$ ahead of $t(i)$ in time). For brevity, we sometimes drop $i$ from $s(i)$ and $t(i)$ in the sequel. In the generic discrete diffusion framework D3PM (Austin et al., 2021), its forward process starts from the clean data distribution $p(\mathbf{x})$ and sequentially corrupts it to the prior distribution $\boldsymbol{\pi}$ with a Markov diffusion kernel $q(\mathbf{z}_t \mid \mathbf{z}_s) = \mathrm{Cat}(\mathbf{z}_t; \mathbf{Q}_{t|s}^\top \mathbf{z}_s)$, where $\mathbf{Q}_{t|s}$ is the state transition matrix with $[\mathbf{Q}_{t|s}]_{ij}$ being the probability that the $i$-th token in the vocabulary transitions to the $j$-th token at time $t$. For example, in (Austin et al., 2021), $\mathbf{Q}_{t|s}$ is defined as:

$$\mathbf{Q}_{t|s} = \alpha_{t|s}\mathbf{I} + (1 - \alpha_{t|s})\mathbf{1}\boldsymbol{\pi}^\top, \tag{1}$$

where $\boldsymbol{\pi}$ is a given prior distribution, $\alpha_{t|s} = \frac{\alpha_t}{\alpha_s}$, with $\alpha_t$ being a pre-defined noise schedule which is strictly decreasing in $t$, $\alpha_0 = 1$ and $\alpha_1 = 0$. We parameterize $\alpha_t = e^{-\sigma(t)}$, where $\sigma(t) : [0,1] \to \mathbb{R}^+$. The following $\sigma(t)$'s have been commonly used: (i) log linear schedule $\sigma(t) = -\log(1 - t)$; (ii) cosine squared schedule $\sigma(t) = -\log \cos^2\left(\frac{\pi}{2}(1-t)\right)$; and (iii) geometric schedule $\sigma(t) = (\sigma_{\min})^{1-t}(\sigma_{\max})^t$, where $\sigma_{\min}$, and $\sigma_{\max}$ are hyperparameters.

The forward process introduces the marginal:

$$q(\mathbf{z}_t \mid \mathbf{x}) = \mathrm{Cat}(\mathbf{z}_t; \mathbf{Q}_t^\top \mathbf{x}), \tag{2}$$

where $\mathbf{Q}_t = \prod_{i=1}^t \mathbf{Q}_{t(i)|s(i)}$. The corresponding posterior is:

$$q(\mathbf{z}_s \mid \mathbf{z}_t, \mathbf{x}) = \mathrm{Cat}\left(\mathbf{z}_s; \frac{\mathbf{Q}_{t|s}\mathbf{z}_t \odot \mathbf{Q}_s^\top \mathbf{x}}{\mathbf{z}_t^\top \mathbf{Q}_t^\top \mathbf{x}}\right), \tag{3}$$

and the reverse process is performed by a parameterized diffusion model $p_\theta$ trained to minimize the Negative Evidence Lower BOund (NELBO):

$$\mathbb{E}_q\left[\underbrace{-\log p_\theta(\mathbf{x}|\mathbf{z}_{t(0)})}_{\mathcal{L}_{\text{recon}}} + \underbrace{\sum_{i=1}^T D_{\mathrm{KL}}\Big(q(\mathbf{z}_{s(i)}|\mathbf{z}_{t(i)}, \mathbf{x})||p_\theta(\mathbf{z}_{s(i)}|\mathbf{z}_{t(i)})\Big)}_{\mathcal{L}_{\text{diffu}}}\right] + \underbrace{D_{\mathrm{KL}}\Big(q(\mathbf{z}_{t(T)} \mid \mathbf{x})||p_\theta(\mathbf{z}_{t(T)})\Big)}_{\mathcal{L}_{\text{prior}}},$$

$$\tag{4}$$

where $D_{\mathrm{KL}}[\cdot]$ denotes the Kullback–Leibler divergence.

### 2.1 UNIFORM DIFFUSION MODEL (UDM)

In the UDM, the uniform diffusion process (Schiff et al., 2024) sets the prior distribution to $\boldsymbol{\pi}_u = \frac{1}{N}$, where the input $\mathbf{x}$ transitions to a randomly chosen token with some probability at each time step. UDLM (Schiff et al., 2024) rewrites the state transition matrix $\mathbf{Q}_{t|s}$ in (1) to:

$$\mathbf{Q}_{t|s} = \alpha_{t|s}\mathbf{I} + (1 - \alpha_{t|s})\mathbf{1}(\frac{1}{N})^\top. \tag{5}$$

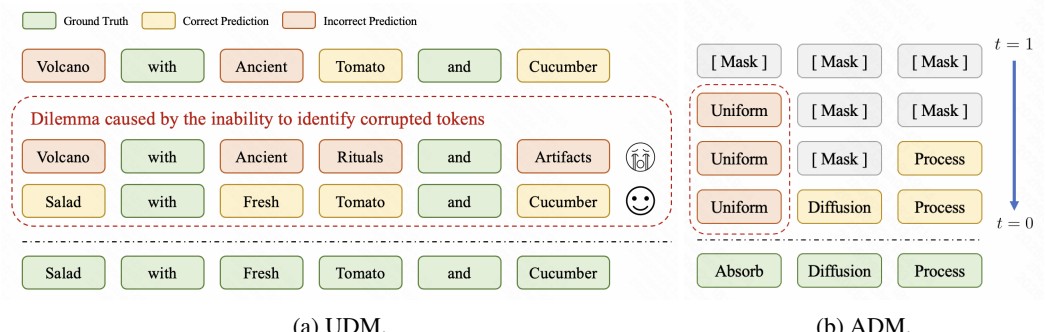

(a) UDM.  (b) ADM.

Figure 1: Examples illustrating the limitations of UDM and ADM.

Importantly, once $\mathbf{x}$ is altered, it may continue to be revised in the subsequent steps. The corresponding posterior distribution in (3) becomes

$$q(\mathbf{z}_s \mid \mathbf{z}_t, \mathbf{x}) = \text{Cat}\left(\mathbf{z}_s; \frac{N\alpha_t \mathbf{z}_t \odot \mathbf{x} + (\alpha_{t|s} - \alpha_t)\mathbf{z}_t + (\alpha_s - \alpha_t)\mathbf{x} + \frac{(\alpha_s - \alpha_t)(1-\alpha_s)}{N\alpha_s}\mathbf{1}}{N\alpha_t \mathbf{z}_t^\top \mathbf{x} + (1 - \alpha_t)}\right). \quad (6)$$

### 2.2 Absorbing Diffusion Model (ADM)

ADMs employ a forward process that gradually corrupts the input sequence $\mathbf{x}^{1:L}$ by replacing tokens with the absorbing state $\mathbf{m}$ (i.e., [MASK] token). At the final time step $T$, all inputs are masked with probability 1. Following MDLM (Sahoo et al., 2024), given the prior distribution to $\boldsymbol{\pi}_m = \mathbf{m}$, the state transition matrix $\mathbf{Q}_{t|s}$ in (1) becomes:

$$\mathbf{Q}_{t|s} = \beta_{t|s}\mathbf{I} + (1 - \beta_{t|s})\mathbf{1}\mathbf{m}^\top. \quad (7)$$

The marginal distribution of the forward process is then $q(\mathbf{z}_t \mid \mathbf{x}) = \text{Cat}(\mathbf{z}_t; \beta_t \mathbf{x} + (1 - \beta_t)\mathbf{m})$. By reparameterization and further derivation, the posterior (3) can be simplified to:

$$q(\mathbf{z}_s \mid \mathbf{z}_t, \mathbf{x}) = \begin{cases} \text{Cat}(\mathbf{z}_s; \mathbf{z}_t) & \mathbf{z}_t \neq \mathbf{m} \\ \text{Cat}\left(\mathbf{z}_s; \frac{(1 - \beta_s)\mathbf{m} + (\beta_s - \beta_t)\mathbf{x}}{1 - \beta_t}\right) & \mathbf{z}_t = \mathbf{m} \end{cases}. \quad (8)$$

Note that for the input unmasked tokens (i.e., $\mathbf{z}_t \neq \mathbf{m}$), ADMs directly copy them to the output. Moreover, DiffuGPT, which adopts ADMs on top of pretrained ARMs, also achieves strong performance.

## 3 Proposed Method

Section 3.1 first discusses the limitations of existing discrete diffusion models. Section 3.2 then introduces the proposed mixture of uniform and absorbing processes.

### 3.1 Limitations of UDM and ADM

**UDMs** corrupt data by transitioning any token to another in the vocabulary, which allows for gradual, token-level refinement. However, UDMs adopt an aggressive noising strategy at the sequence level, as shown in (5), which often induces semantic shifts and distorts the sequence's meaning. Therefore, during the reverse process, the model must first identify the corrupted tokens before attempting to denoise them, which increases prediction difficulty (Gu et al., 2022). Moreover, semantic conflicts can arise across local contexts, causing competing token predictions and making it unclear which positions are reliable (Amin et al., 2025; Gu et al., 2022). For example, as illustrated in Figure 1a, the model might incorrectly identify and revise corrupted tokens, resulting in an erroneous output.

**ADMs** perform gradual denoising at the sequence level. However, as shown in (8), at the token level its reverse denoising process directly copies the input unmasked tokens to the output (the *Carry-Over*

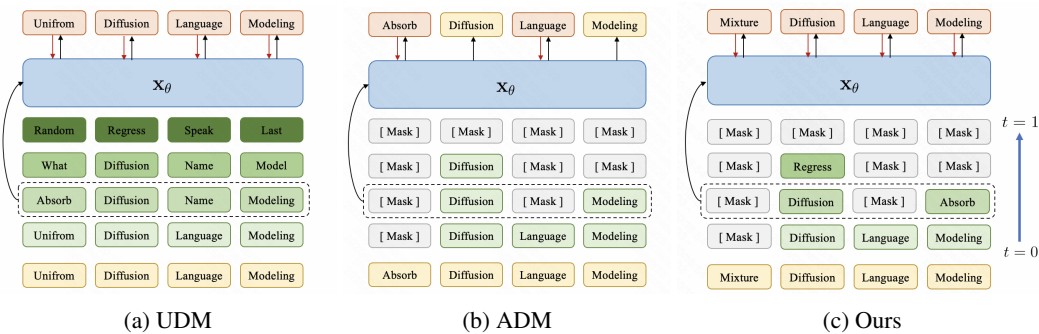

(a) UDM         (b) ADM         (c) Ours

Figure 2: **Method Overview.** (a) UDM: the noising process replaces tokens with random tokens. (b) ADM: the noising process replaces tokens with [MASK] token. (c) Proposed MAUD: the noising process replaces tokens with a mixture of [MASK] token and random tokens.

*Unmasked Tokens* property (Sahoo et al., 2024)). This induces a biased assumption that unmasked tokens are inherently correct, causing the model to lose its ability to further correct. Clearly, this is inconsistent with the fundamental idea of refinement in diffusion processes. For example, as illustrated in Figure 1b, if the model initially predicts the incorrect token "Uniform", it cannot later revise it to the correct token "Absorbing". The performance improvements observed in approaches incorporating test-time optimization (Peng et al., 2025; Kim et al., 2025) or explicit self-correction mechanisms (Liu et al., 2025) provide further evidence that this issue is fundamental.

The distinct limitations of ADMs and UDMs suggest that they are highly complementary. UDMs preserve the gradual essence of diffusion at the token level, but their sequence-level changes are overly drastic. In contrast, ADMs produce very smooth changes at the sequence level, yet are overly conservative at the token level, as only a single type of transitions is permitted. This naturally raises the following question:

> *How to design a unified process that leverages the strengths of both processes?*

### 3.2 MIXTURE OF ABSORBING AND UNIFORM DIFFUSION

To efficiently leverage the strengths of these two processes while addressing their limitations, we propose Mixture of Absorbing and Uniform Diffusion (MAUD). We first introduce its forward and reverse processes in Sections 3.2.1 and 3.2.2, respectively, and then derive the likelihood bounds in Section 3.2.3. Finally, Section 3.2.4 describes the training and sampling algorithms. An overview of the proposed method is shown in Figure 2.

#### 3.2.1 FORWARD PROCESS

We introduce two noise schedules, $\alpha_t$ and $\beta_t$, corresponding to the uniform and absorbing diffusion processes, respectively. The prior distribution is defined as follows. For the masked diffusion process, its prior is $\pi_m = \mathbf{m}$. For the uniform diffusion process, the prior is $\pi_u = \frac{\mathbf{u}}{N}$, where $\mathbf{u} = \mathbf{1} - \mathbf{m}$. The state transition matrix, $\mathbf{Q}_{t|s}$, is then given by:

$$\mathbf{Q}_{t|s} = \underbrace{(1 - \beta_{t|s})\mathbf{1}\mathbf{m}^\top}_{\text{Absorbing Diffusion}} + \beta_{t|s}\underbrace{\left[(1 - \alpha_{t|s})\mathbf{u}\frac{\mathbf{u}^\top}{N} + (1 - \alpha_{t|s})\mathbf{m}\mathbf{m}^\top + \alpha_{t|s}\mathbf{I}\right]}_{\text{Uniform Diffusion}}.$$

In other words, at the diffusion step from $s \to t$, a fraction $(1 - \beta_{t|s})$ of the probability mass is transferred to the absorbing process's prior distribution $\mathbf{m}$, while the remaining $\beta_{t|s}$ is uniformly diffused over $\mathcal{V}$ excluding the [MASK] token. Similarly, a fraction $(1 - \alpha_{t|s})$ of the probability mass is redistributed uniformly across the non-masked tokens in $\mathcal{V}$, while the remaining $\alpha_{t|s}$ is retained on the original token through the identity mapping.

Due to the property of the Markov chain, one can marginalize over the intermediate steps and derive the probability of $\mathbf{z}_t$ given $\mathbf{x}$ as $q(\mathbf{z}_t \mid \mathbf{x}) = \mathrm{Cat}(\mathbf{z}_t; \mathbf{Q}_t^\top \mathbf{x})$ (see Appendix B for details).

### 3.2.2 Reverse Denoising Process

Using the properties of diffusion process, the posterior $q(\mathbf{z}_s \mid \mathbf{z}_t, \mathbf{x})$ can be simplified as

$$q(\mathbf{z}_s \mid \mathbf{z}_t, \mathbf{x}) = \begin{cases} \mathrm{Cat}\left(\mathbf{z}_s; \dfrac{(\beta_s - \beta_t)\left[\alpha_s \mathbf{x} + (1 - \alpha_s)\frac{\mathbf{u}}{N}\right] + (1 - \beta_s)\mathbf{m}}{1 - \beta_t}\right) & \mathbf{z}_t = \mathbf{m}, \\ \mathrm{Cat}\left(\mathbf{z}_s; \dfrac{N\alpha_t \mathbf{z}_t \odot \mathbf{x} + (\alpha_{t|s} - \alpha_t)\mathbf{z}_t + (\alpha_s - \alpha_t)\mathbf{x} + \frac{(\alpha_s - \alpha_t)(1 - \alpha_s)}{N\alpha_s}\mathbf{u}}{N\alpha_t \mathbf{z}_t^\top \mathbf{x} + (1 - \alpha_t)}\right) & \mathbf{z}_t \neq \mathbf{m}. \end{cases}$$

$$(9)$$

The reverse process can be described as follows: If $\mathbf{z}_t = \mathbf{m}$, $\mathbf{z}_s$ either remains at $\mathbf{m}$ with probability $\frac{1-\beta_s}{1-\beta_t}$ or transitions to the uniformly corrupted distribution $\alpha_s \mathbf{x} + (1 - \alpha_s)\frac{\mathbf{u}}{N}$ with probability $\frac{\beta_s - \beta_t}{1-\beta_t}$. This differs from ADM in (8), which instead remains at $\mathbf{m}$ with probability $\frac{1-\beta_s}{1-\beta_t}$ or transitions to the one-hot distribution $\mathbf{x}$ with probability $\frac{\beta_s - \beta_t}{1-\beta_t}$. If $\mathbf{z}_t \neq \mathbf{m}$, the reverse process reduces to a pure uniform reverse process, equivalent to that of UDMs in (6).

The optimal form of the reverse diffusion process $p_\theta(\mathbf{z}_s \mid \mathbf{z}_t)$ is given by (9). However, setting $p_\theta$ exactly equal to that in (9) is infeasible, as it cannot directly depend on $\mathbf{x}$. To address this, as in MDLM (Sahoo et al., 2024), we introduce a network $\mathbf{x}_\theta(\mathbf{z}_t, t) : \mathcal{V} \times [0, 1] \to \Delta^N$ that approximates the clean data $\mathbf{x}$ from the noisy latent $\mathbf{z}_t$.

Notably, by introducing uniform reverse process after the absorbing reverse process, we naturally eliminate the *Carry-Over Unmasked Tokens* property. Furthermore, this design choice allows MAUD to simultaneously preserve semantic gradualness at the sequence level through the absorbing process and enable effective error correction via the uniform diffusion process, thereby combining the advantages of both approaches while mitigating their individual limitations.

### 3.2.3 Likelihood Bounds

It has been shown empirically and mathematically that increasing the number of steps $T$ yields a tighter approximation to the NELBO (Song & Ermon, 2019; Sahoo et al., 2024; Schiff et al., 2024). Following this, we develop an NELBO by taking $T \to \infty$ and analyze each of the terms in (4).

**Prior Loss $\mathcal{L}_{\mathbf{prior}}$.** If the noise schedule satisfying $\lim_{T \to \infty} \beta_{t(T)} = 0$, we have $\lim_{T \to \infty} \mathbf{Q}_{t(T)} = \mathbf{1m}^\top$. Consequently, $\lim_{T \to \infty} q(\mathbf{z}_{t(T)} \mid \mathbf{x}) = \lim_{T \to \infty} \mathrm{Cat}(\mathbf{z}_{t(T)}; \mathbf{Q}_{t(T)}\mathbf{x}) = \boldsymbol{\pi}_m$. By setting $p_\theta(\mathbf{z}_{t(T)}) = \boldsymbol{\pi}_m$, the KL divergence in $\mathcal{L}_{\mathrm{prior}}$ becomes zero.

**Diffusion Loss $\mathcal{L}_{\mathbf{diffu}}$.** Based on (9), the diffusion loss is calculated based on whether the token is masked or unmasked. For convenience, we let $\mathbf{x}_j$ be the $j$-th index of a vector $\mathbf{x}$, $\mathcal{F}(\mathbf{x}, \alpha_t) = N\alpha_t \mathbf{x} + (1 - \alpha_t)\mathbf{1}$, and $i = \arg\max_{j \in [N]}(\mathbf{z}_t)_j$ be the largest nonzero entry of $\mathbf{z}_t$. For the masked tokens ($\mathbf{z}_t = \mathbf{m}$), we have (see Appendix B.4 for details):

$$\mathcal{L}_{\mathrm{diffu}}^m = \mathbb{E}_{q,t}\left[\frac{\beta_t'}{N(1 - \beta_t)}\left(\mathcal{F}(\mathbf{1}, \alpha_s)_i \log \frac{\mathcal{F}(\mathbf{x}_\theta, \alpha_s)_i}{\mathcal{F}(\mathbf{1}, \alpha_s)_i} + \sum_{j \neq i}(1 - \alpha_s)\log \frac{\mathcal{F}(\mathbf{x}_\theta, \alpha_s)_j}{1 - \alpha_s}\right)\right].$$

For the unmasked tokens ($\mathbf{z}_t \neq \mathbf{m}$) we expand the $\mathcal{L}_{\mathrm{diffu}}$ as follows (see Appendix B.4 for details):

$$\mathcal{L}_{\mathrm{diffu}}^u = \mathbb{E}_{q,t}\left[\frac{\alpha_t'}{N\alpha_t}\left[\frac{N}{\mathcal{F}(\mathbf{x}, \alpha_t)_i} - \frac{N}{\mathcal{F}(\mathbf{x}_\theta, \alpha_t)_i} - \sum_{j \neq i}\frac{\mathcal{F}(\mathbf{x}, \alpha_t)_j}{\mathcal{F}(\mathbf{x}, \alpha_t)_i}\log\left(\frac{\mathcal{F}(\mathbf{x}_\theta, \alpha_t)_i}{\mathcal{F}(\mathbf{x}_\theta, \alpha_t)_j} \cdot \frac{\mathcal{F}(\mathbf{x}, \alpha_t)_j}{\mathcal{F}(\mathbf{x}, \alpha_t)_i}\right)\right]\right].$$

The total diffusion loss combines these two cases with the Kronecker delta function $\delta_{\mathbf{z}_t, \mathbf{m}}$

$$\mathcal{L}_{\mathrm{diffu}} = \delta_{\mathbf{z}_t, \mathbf{m}}\mathcal{L}_{\mathrm{diffu}}^m + (1 - \delta_{\mathbf{z}_t, \mathbf{m}})\mathcal{L}_{\mathrm{diffu}}^u.$$

**Reconstruction Loss $\mathcal{L}_{\textbf{recon}}$.** Given noise schedule that satisfies $\lim_{T\to\infty} \alpha_{t(\frac{1}{T})} = 1$ and $\lim_{T\to\infty} \beta_{t(\frac{1}{T})} = 1$, we have $\lim_{T\to\infty} \mathbf{Q}_{t(\frac{1}{T})} = \mathbf{I}$, Consequently, the marginal distribution $q(\mathbf{z}_{t(\frac{1}{T})} \mid \mathbf{x})$ becomes $\lim_{T\to\infty} q(\mathbf{z}_{t(\frac{1}{T})} \mid \mathbf{x}) = \text{Cat}(\mathbf{z}_{t(\frac{1}{T})}; \mathbf{x})$. In other words, in the limit, the first latent variable is exactly equal to the clean data. Thus, in the continuous-time limit, we have $\mathcal{L}_{\text{recon}} = \lim_{T\to\infty} \mathbb{E}_{q(\mathbf{z}_{t(\frac{1}{T})}|\mathbf{x})} \log p_\theta(\mathbf{x} \mid \mathbf{z}_{t(\frac{1}{T})}) = 0$.

**Extension to Sequences.** To extend training from $\mathbf{x} \in \mathcal{V}$ to sequences $\mathbf{x}^{1:L} \in \mathcal{V}^L$, following Sahoo et al. (2024), we make the assumption that the denoising process factorizes independently across tokens when conditioned on a sequence of noisy latents $\mathbf{z}_t^{1:L}$. In this case, we use a single model $\mathbf{x}_\theta^\ell(\mathbf{z}_t^{1:L}, t)$ for predicting each token $\ell \in \{1, \dots, L\}$ in a sequence.

### 3.2.4 Training and Sampling

**Training.** During training, we observe that different choices of noise schedulers affect the likelihood bounds of MAUD. For the absorbing noise scheduler $\beta_t$, it can be shown that the diffusion loss $\mathcal{L}_{\text{diffu}}^u$ is invariant to its functional form (see Appendix C for details), which is consistent with Sahoo et al. (2024). We therefore adopt the log-linear schedule following the standard practice (Lou et al., 2024). In contrast, for the uniform noise scheduler $\alpha_t$, we observe that a geometric scheduler yields significantly better results than a log-linear one. Detailed experiments and analysis are provided in Section 4.4.

---

**Algorithm 1** UMDM Sampling.

---

1: **Input:** Network $\mathbf{x}_\theta$, sampling steps $T$, masked noise scheduler $\beta$, uniform noise scheduler $\alpha$
2: **Initialize:** $t \leftarrow 1$, $\Delta t = \frac{1}{T}$ $\mathbf{z}_t^{1:L} \sim \{\mathbf{m}\}^L$
3: **while** $t > 0$ **do**
4:     $s \leftarrow t - \Delta t$
5:     Estimate clean data $\hat{\mathbf{x}}^\ell = \mathbf{x}_\theta^\ell(\mathbf{z}_t^{1:L}, t)$
6:     $\forall \mathbf{z}_t^\ell = \mathbf{m}, p_\theta^m(\mathbf{z}_s^\ell \mid \mathbf{z}_t^{1:L}) = \dfrac{(\beta_s - \beta_t)\left[\alpha_s \hat{\mathbf{x}}^\ell + (1 - \alpha_s)\frac{\mathbf{u}}{N}\right] + (1 - \beta_s)\mathbf{m}}{1 - \beta_t}$
7:     $\forall \mathbf{z}_t^\ell \neq \mathbf{m}, p_\theta^u(\mathbf{z}_s^\ell \mid \mathbf{z}_t^{1:L}) = \dfrac{N\alpha_t \mathbf{z}_t^\ell \odot \hat{\mathbf{x}}^\ell + (\alpha_{t|s} - \alpha_t)\mathbf{z}_t^\ell + (\alpha_s - \alpha_t)\hat{\mathbf{x}}^\ell + \frac{(\alpha_s - \alpha_t)(1 - \alpha_s)}{N\alpha_s}\mathbf{u}}{N\alpha_t \mathbf{z}_t^{\top}\hat{\mathbf{x}}^\ell + (1 - \alpha_t)}$
8:     $\forall \mathbf{z}_t^\ell = \mathbf{m}, \mathbf{z}_s^\ell \sim \text{Cat}(\mathbf{z}_s^\ell; p_\theta^m(\mathbf{z}_s^\ell \mid \mathbf{z}_t^{1:L})), \forall \mathbf{z}_t^\ell \neq \mathbf{m}, \mathbf{z}_s^\ell \sim \text{Cat}(\mathbf{z}_s^\ell; p_\theta^u(\mathbf{z}_s^\ell \mid \mathbf{z}_t^{1:L}))$
9:     $\mathbf{z}_t^{1:L} \leftarrow \mathbf{z}_s^{1:L}$
10:    $t \leftarrow t - \Delta t$
11: **end while**
12: **Return:** $\mathbf{z}_t^{1:L}$

---

**Sampling.** Following Sahoo et al. (2024), we initialize $\mathbf{z}_{t(T)}^{1:L}$ with all [MASK] tokens and then sample tokens according to the posterior $q(\mathbf{z}_s^\ell \mid \mathbf{z}_t^\ell, \mathbf{x})$ in (9). At each timestep, if $\mathbf{z}_t^\ell$ is a [MASK] token, it transitions to the predicted unmasked token at time $s$ with probability $\frac{\beta_s - \beta_t}{1 - \beta_t}$ (Algo. 1, line 6). If $\mathbf{z}_t^\ell$ is an unmasked token, it undergoes a uniform denoising step (Algo. 1, line 7). This process continues for $T$ steps to generate the final sequence.

## 4 Experiments

In this section, we evaluate MAUD on both language generation and understanding tasks.

### 4.1 Setup

**Datasets.** For the language generation task, we use two widely used benchmark datasets: (i) Text8 (Mahoney, 2006), a relatively small dataset designed for character-level modeling; and (ii) OpenWebText (Gokaslan & Cohen, 2019), an open-source replication of the unpublished WebText corpus. For the language understanding task, we evaluate on a suite of challenging benchmarks

spanning commonsense reasoning and mathematical problem solving. Specifically, we use Trivi-aQA (Joshi et al., 2017) to test the reading comprehension of models. We also include commonsense reasoning tasks HSwag (Zellers et al., 2019), Wino (Sakaguchi et al., 2021), SIQA (Sap et al., 2019), and PIQA (Bisk et al., 2020), all of which involve multiple-choice questions. On grade school math problems GSM8K (Cobbe et al., 2021), we follow Ye et al. (2024) in the finetuning setting using the augmented symbolic data to test the CoT (Wei et al., 2022) math reasoning abilities.

**Baselines.** We compare against three categories of baselines: (i) classical autoregressive models, including GPT2 (Brown et al., 2020) and LLaMA (Touvron et al., 2023); (ii) continuous diffusion models, including Plaid-1B (Gulrajani & Hashimoto, 2023) and Bayesian Flow Network (BFN) (Graves et al., 2023); and (iii) state-of-the-art discrete diffusion models, including the generic DDM models D3PM (Austin et al., 2021) and SEDD (Lou et al., 2024) (both applicable as UDMs or ADMs), the classical ADM model MDLM[1] (Sahoo et al., 2024), and the recent UDM implementation UDLM (Schiff et al., 2024). For D3PM and SEDD, we denote their uniform and absorbing variants by (Uni.) and (Abs.), respectively. On the Text8 dataset, we include other discrete sequence generation models for comparison, including flow-based method IAF/SCF (Ziegler & Rush, 2019), Argmax Flows (Hoogeboom et al., 2021), and Discrete Flows (Tran et al., 2019). For language understanding tasks, we also use DiffuGPT (Gong et al., 2024) as baseline.

**Metrics.** For the language generation task, following standard practice (Lou et al., 2024), we report Bits Per Character (BPC) and Perplexity (PPL). Both BPC and PPL quantify the model's predictive uncertainty over the true test data, and thus evaluate its ability to assign likelihoods to the observed sequences. Language understanding tasks are evaluated using task-specific metrics. Following Gong et al. (2024), TriviaQA is measured by the exact match accuracy. For commonsense reasoning benchmarks formulated as multiple-choice questions, we report the answer accuracy. Finally, performance on GSM8K is evaluated based on the correctness of the predicted solutions to mathematical word problems.

More experimental details are provided in Appendix D.

| Method | BPC (↓) |
|---|---|
| *Autoregressive* | |
| IAF/SCF | 1.88 |
| Argmax Flows | 1.39 |
| Discrete Flows | **1.23** |
| Transformer | **1.23** |
| *Continuous Diffusion* | |
| Plaid | 1.48 |
| BFN | 1.41 |
| *Discrete Diffusion* | |
| D3PM[†] | 1.45 |
| SEDD (Uni.)[†] | 1.47 |
| SEDD (Abs.)[†] | 1.39 |
| MDLM | 1.38 |
| UDLM[†] | 1.44 |
| **MAUD (*Ours*)** | 1.36 |

Table 1: Bits Per Character (BPC) on Text8 test set. [†] denotes the numbers are borrowed from Schiff et al. (2024).

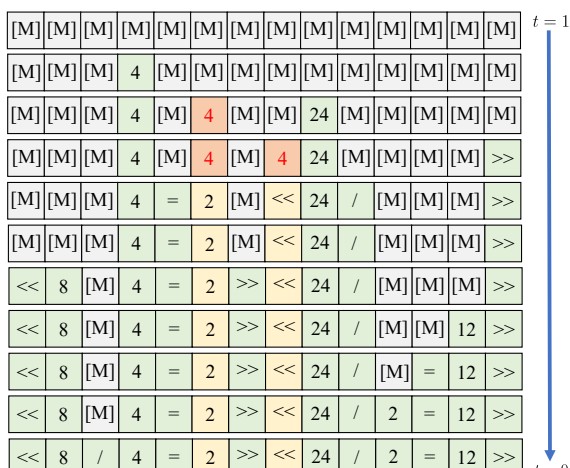

Figure 3: Visualization of MAUD's mathematical problem-solving process.

## 4.2 LANGUAGE GENERATION

**Text8.** Following Schiff et al. (2024), we report the BPC in Table 1. As can be seen, MAUD achieves substantial improvements over flow-based autoregressive models IAF/SCF and Argmax Flows, and is outperformed only by the autoregressive Transformers and Discrete Flows that in-

---

[1]This is reimplemented and trained under the same experimental settings as MAUD.

| Method | Wiki | PTB | 1BW | Lambada | News | Pubmed | Arxiv |
|---|---|---|---|---|---|---|---|
| *Autoregressive* | | | | | | | |
| GPT2* | **25.75** | **82.05** | **51.25** | 51.28 | **52.09** | 49.01 | 41.73 |
| *Continuous Diffusion* | | | | | | | |
| Plaid[†] | 50.86 | 142.60 | 91.12 | 57.28 | - | - | - |
| *Discrete Diffusion* | | | | | | | |
| D3PM[†] | 75.16 | 168.27 | 138.92 | 93.47 | - | - | - |
| SEDD (Uni.)[†] | 44.12 | 124.07 | 89.96 | 63.43 | 71.24 | 50.79 | 44.21 |
| SEDD (Abs.)[†] | 34.28 | 100.09 | 68.20 | 49.86 | 62.09 | 44.53 | 38.48 |
| MDLM | 32.83 | 95.26 | 67.01 | 47.52 | 61.15 | 41.89 | **37.37** |
| **MAUD (*Ours*)** | 31.94 | 92.82 | 66.10 | **46.43** | 59.85 | **40.17** | 37.41 |

Table 2: Zero-shot unconditional perplexity ($\downarrow$) across various datasets. [†] denotes the numbers are borrowed from Sahoo et al. (2024). *The GPT2 numbers are reported for the checkpoint pretrained on WebText instead of OpenWebText and thus is not a direct comparison.

| Model | Param | QA | Math | Common Reasoning | | | |
|---|---|---|---|---|---|---|---|
| | | TriQA | GSM8K | HSwag | Wino. | SIQA | PIQA |
| *Autoregressive* | | | | | | | |
| GPT2 | 127M | 4.0 | 44.8 | 29.9 | 48.5 | 35.7 | 62.1 |
| LLaMA | 7.0B | **45.4** | **58.6** | **74.9** | **67.1** | **44.8** | **78.3** |
| *Continuous Diffusion* | | | | | | | |
| Plaid | 1.3B | 1.2 | 32.6 | 39.3 | 51.3 | 32.3 | 54.5 |
| *Discrete Diffusion* | | | | | | | |
| SEDD (Abs.) | 170M | 1.5 | 45.3 | 30.2 | 50.1 | 34.4 | 55.6 |
| MDLM | 127M | 1.8 | 47.8 | 31.1 | 50.3 | 36.4 | 55.8 |
| DiffGPT | 127M | 2.0 | 50.2 | 33.4 | 50.8 | 37.0 | 57.7 |
| **MAUD (*Ours*)** | 127M | 1.9 | 51.4 | 32.8 | 51.0 | 37.3 | 56.5 |

Table 3: Performance ($\uparrow$) on language understanding benchmarks. We finetune models on GSM8K; other datasets are all in zero-shot setting.

corporate autoregressive-based distributions. Moreover, MAUD consistently outperforms existing UDMs and MDMs, establishing a new state-of-the-art among diffusion-based language models.

**OpenWebText.** Following Sahoo et al. (2024), we train models on OpenWebText and evaluate the perplexity on a diverse suite of benchmarks, including WikiText (Merity et al., 2017), PTB (Marcus et al., 1993), 1BW (Chelba et al., 2013), Lambada (Paperno et al., 2016), AG News (Zhang et al., 2015), and the PubMed and ArXiv subsets of Scientific Papers (Cohan et al., 2018). Results are shown in Table 2. Compared with UDMs such as SEDD (Uni.), MAUD yields substantial performance improvements across all test sets. Relative to ADMs, including D3PM, SEDD(Abs.), and MDLM, MAUD attains lower perplexity on the majority of benchmarks. Furthermore, among all diffusion-based approaches, MAUD establishes a new state-of-the-art, narrowing the performance gap to autoregressive models more than any prior method.

### 4.3 LANGUAGE UNDERSTANDING

Following Gong et al. (2024), we fine-tune the OpenWebText-pretrained model on GSM8K, adopting the fine-tuning setup strictly as described in Ye et al. (2024). The fine-tuned model is then used for language understanding evaluations. As shown in Table 3, MAUD is the state-of-the-art across a broad set of language understanding benchmarks among models with similar parameter budget. Its slightly lower performance on PIQA relative to GPT-2 may be attributed to the task's reliance on specific physical knowledge, which our models may lack. This limitation likely stems from fine-tuning on models trained with only ~30B tokens of OpenWebText, a scale that may be

insufficient for acquiring broad physical commonsense knowledge. In contrast, on tasks requiring extensive global reasoning, such as GSM8K, MAUD consistently outperforms GPT2 that rely solely on left-to-right modeling.

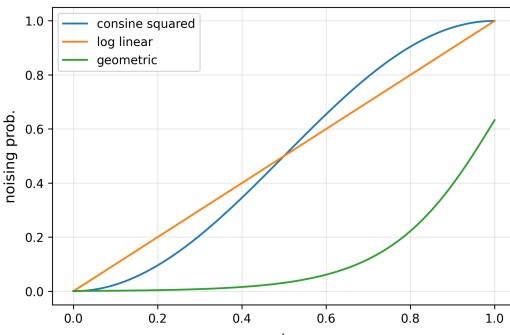

Figure 4: Illustration of how the noising probability varies with timestep $t$ under different schedules. For gemetric schedule, we set $\sigma_{min} = 1 \times 10^{-3}$ and $\sigma_{\max} = 1$.

| $\alpha_t$ | Metric |
|---|---|
| *Text8* | *BPC ($\downarrow$)* |
| Log Linear | 1.38 |
| Cosine Squared | 1.40 |
| Geometric ($\sigma_{\max} = 1$) | **1.36** |
| *Openwebtext* | *Perplexity ($\downarrow$)* |
| Log Linear | 23.64 |
| Geometric ($\sigma_{\max} = 1$) | **22.98** |

Table 4: Effect of uniform noise schedule on MAUD's performance. We report BPC on Text8 and perplexity on OpenWebText.

Figure 3 shows an illustration of MAUD's reverse generation process, given the problem *"A sloth takes 4 hours to go down, collect berries, and return to its tree. In 8 hours, it wants to collect 24 berries. What is the minimum number of berries it must pick per trip?"* . As can be seen, while MAUD may generate incorrect tokens (red) during the absorbing reverse phase, they can be subsequently corrected in the uniform reverse phase (yellow).

### 4.4 ABLATION STUDY: EFFECT OF UNIFORM NOISE SCHEDULER

In this experiment, we use the Text8 and OpenWebText test sets to investigate how different choices of the uniform noise scheduler $\alpha_t$ influence the likelihood bounds of MAUD. As for the absorbing noise scheduler $\beta_t$, we follow standard practice (Lou et al., 2024) and use the log-linear schedule. We illustrate in Figure 4 how the noising probability varies with timestep $t$ under different schedules.

As can be seen from Table 4, the log-linear schedule performs poorly. We attribute to its excessively high uniform noise probability during the early stages of generation. This overrides the relatively stable prior knowledge obtained from the absorbing reverse process, causing the model to nearly degenerate into a pure UDM. In contrast, the geometric scheduler assigns lower transition probabilities to unmasked tokens in the early stages of generation, thereby preserving more of the prior knowledge carried by the absorbing states. In the later stages, once sufficient prior knowledge has been accumulated, transition probabilities for the unmasked tokens naturally decay to $0$. This ensures that MDMs can perform denoising effectively without requiring additional corrections from UDMs. This scheduling strategy balances early preservation of prior knowledge with late-stage stability, explaining the superior performance of the geometric scheduler.

## 5 CONCLUSION

In this paper, we introduced MAUD, a novel discrete diffusion model that interpolates between absorbing and uniform diffusion processes. By combining the token-level refinement of uniform diffusion with the semantic stability of absorbing diffusion, MAUD overcomes the key limitations of prior methods and achieves state-of-the-art experimental results. Future work includes applying MAUD to other discrete domains, such as protein modeling, and further scaling our model to larger datasets and parameter counts. At the same time, developing more effective ways to define a gradual diffusion process in the discrete domain remains a highly promising direction for future research.

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

## A  RELATED WORKS

**Discrete Diffusion Models.**  Continuous diffusion models have demonstrated remarkable performance and controllability in image generation tasks (Ho et al., 2020; Song & Ermon, 2019; Rombach et al., 2022; Lipman et al., 2023; Esser et al., 2024). Building on these successes, several works have extended continuous diffusion to text generation (Gong et al., 2023; Gulrajani & Hashimoto, 2023). Among them, Plaid (Gulrajani & Hashimoto, 2023) is a notable approach that maps discrete text into a continuous embedding space and performs diffusion in that space. Given the inherently discrete nature of language, Austin et al. (2021) proposed D3PM, a generic discrete diffusion framework tailored to discrete data. Lou et al. (2024) further introduced score matching into the discrete domain, yielding a tighter ELBO. Depending on the noising mechanism, discrete diffusion models can be broadly categorized into two types. The first is the Uniform Diffusion Model (UDM), which applies noise by randomly replacing a token with another token from the vocabulary; the framework of Schiff et al. (2024) is the most widely used instantiation of this class. The second is the Absorbing Diffusion Model (ADM), which introduces an absorbing state `[MASK]` and progressively replaces tokens with `[MASK]` to apply noise, with MDLM (Sahoo et al., 2024) serving as the canonical framework. By scaling up MDLM, Dream-7B (Ye et al., 2025b) and LLaDA-8B (Nie et al., 2025) demonstrate language understanding performance comparable to large autoregressive language models. Furthermore, Ye et al. (2025a) show that ADMs substantially outperform autoregressive models on tasks requiring complex reasoning and global planning.

## B  MIXTURE OF ABSORBING AND UNIFROM DIFFUSION

Given the noise schedules $\alpha_t$ and $\beta_t$ for uniform and absorbing diffusion process, respectively. We specify the prior distributions as follows. For the absorbing diffusion process, following Sahoo et al. (2024), the prior distribution is given by $\boldsymbol{\pi}_m = \mathbf{m}$. For uniform noise diffusion, we define the prior distribution as $\boldsymbol{\pi}_u = \frac{\mathbf{u}}{N}$, where $\mathbf{u} = \mathbf{1} - \mathbf{m}$.

Thus, the state transition matrix $\mathbf{Q}_{t|s}$ is defined as

$$\mathbf{Q}_{t|s} = \beta_{t|s} \left[ \alpha_{t|s}\mathbf{I} + (1 - \alpha_{t|s})\mathbf{m}\mathbf{m}^\top + (1 - \alpha_{t|s})\mathbf{u}\frac{\mathbf{u}^\top}{N} \right] + (1 - \beta_{t|s})\mathbf{1}\mathbf{m}^\top \tag{10}$$

where $\alpha_{t|s} = \frac{\alpha_t}{\alpha_s}$. Then, we have:

$$\mathbf{Q}_t = \prod_{i=1}^{t} \mathbf{Q}_{t(i)|s(i)} = \beta_t \left[ \alpha_t\mathbf{I} + (1 - \alpha_t)\mathbf{m}\mathbf{m}^\top + (1 - \alpha_t)\mathbf{u}\frac{\mathbf{u}^\top}{N} \right] + (1 - \beta_t)\mathbf{1}\mathbf{m}^\top \tag{11}$$

### B.1  FORWARD PROCESS

Accordingly, the forward process of MAUD can be written as follows:

$$
\begin{aligned}
&q(\mathbf{z}_t \mid \mathbf{z}_s) \\
&= \text{Cat}\left( \mathbf{z}_t; \mathbf{Q}_{t|s}^\top \mathbf{z}_s \right) \\
&= \text{Cat}\left( \mathbf{z}_t; \left[ \beta_{t|s}\left( \alpha_{t|s}\mathbf{I} + (1 - \alpha_{t|s})\mathbf{m}\mathbf{m}^\top + (1 - \alpha_{t|s})\mathbf{u}\frac{\mathbf{u}^\top}{N} \right) + (1 - \beta_{t|s})\mathbf{m}\mathbf{1}^\top \right] \mathbf{z}_s \right)
\end{aligned}
\tag{12}
$$

### B.2  REVERSE PROCESS

Consider the case $\mathbf{z}_t = \mathbf{m}$, i.e. $\mathbf{z}_t$ is unmasked, $q(\mathbf{z}_s \mid \mathbf{z}_t, \mathbf{x})$ simplifies to the following:

$$
\begin{aligned}
&q(\mathbf{z}_s \mid \mathbf{z}_t = \mathbf{m}, \mathbf{x}) \\
&= \text{Cat}\left( \mathbf{z}_s; \frac{\left[ \beta_{t|s}\mathbf{m} + (1 - \beta_{t|s})\mathbf{1} \right] \odot \left[ \beta_s\left( \alpha_s\mathbf{x} + (1 - \alpha_s)\frac{\mathbf{u}}{N} \right) + (1 - \beta_s)\mathbf{m} \right]}{1 - \beta_t} \right) \\
&= \text{Cat}\left( \mathbf{z}_s; \frac{(\beta_s - \beta_t)\left[ \alpha_s\mathbf{x} + (1 - \alpha_s)\frac{\mathbf{u}}{N} \right] + (1 - \beta_s)\mathbf{m}}{1 - \beta_t} \right)
\end{aligned}
\tag{13}
$$

Consider the case $\mathbf{z}_t \neq \mathbf{m}$, i.e. $\mathbf{z}_t$ is masked, $q(\mathbf{z}_s \mid \mathbf{z}_t, \mathbf{x})$ simplifies to the following:

$$
\begin{aligned}
& q(\mathbf{z}_s \mid \mathbf{z}_t \neq \mathbf{m}, \mathbf{x}) \\
& = \mathrm{Cat}\left(\mathbf{z}_s; \frac{\beta_{t|s}\left[\alpha_{t|s}\mathbf{z}_t + (1 - \alpha_{t|s})\frac{\mathbf{u}}{N}\right] \odot \left[\beta_s\left(\alpha_s\mathbf{x} + (1 - \alpha_s)\frac{\mathbf{u}}{N}\right) + (1 - \beta_s)\mathbf{m}\right]}{\beta_t\left[\alpha_t\mathbf{z}_t^\top\mathbf{x} + (1 - \alpha_t)\frac{1}{N}\right]}\right) \\
& = \mathrm{Cat}\left(\mathbf{z}_s; \frac{N\alpha_t\mathbf{z}_t \odot \mathbf{x} + (\alpha_{t|s} - \alpha_t)\mathbf{z}_t + (\alpha_s - \alpha_t)\mathbf{x} + \frac{(\alpha_s - \alpha_t)(1 - \alpha_s)}{N\alpha_s}\mathbf{u}}{N\alpha_t\mathbf{z}_t^\top\mathbf{x} + (1 - \alpha_t)}\right)
\end{aligned}
\tag{14}
$$

## B.3 APPROXIMATION OF REVERSE PROCESS

Following Sahoo et al. (2024), we induce a key property into the denoising model $\mathbf{x}_\theta$: we design $\mathbf{x}_\theta$ such that $\mathbf{m}^\top\mathbf{x}_\theta = 0$, i.e., we substitute the logit corresponding to the [MASK] token with $-\infty$.

Consider the case $\mathbf{z}_t = \mathbf{m}$, $p_\theta(\mathbf{z}_s \mid \mathbf{z}_t)$ simplifies to the following:

$$
\begin{aligned}
& q(\mathbf{z}_s \mid \mathbf{z}_t = \mathbf{m}, \mathbf{x}) \\
& = \mathrm{Cat}\left(\mathbf{z}_s; \frac{\left[\beta_{t|s}\mathbf{m} + (1 - \beta_{t|s})\mathbf{1}\right] \odot \left[\beta_s\left(\alpha_s\mathbf{x}_\theta + (1 - \alpha_s)\frac{\mathbf{u}}{N}\right) + (1 - \beta_s)\mathbf{m}\right]}{1 - \beta_t}\right) \\
& = \mathrm{Cat}\left(\mathbf{z}_s; \frac{(\beta_s - \beta_t)\left[\alpha_s\mathbf{x}_\theta + (1 - \alpha_s)\frac{\mathbf{u}}{N}\right] + (1 - \beta_s)\mathbf{m}}{1 - \beta_t}\right)
\end{aligned}
\tag{15}
$$

Consider the case $\mathbf{z}_t \neq \mathbf{m}$, $p_\theta(\mathbf{z}_s \mid \mathbf{z}_t)$ simplifies to the following:

$$
\begin{aligned}
& q(\mathbf{z}_s \mid \mathbf{z}_t \neq \mathbf{m}, \mathbf{x}) \\
& = \mathrm{Cat}\left(\mathbf{z}_s; \frac{\beta_{t|s}\left[\alpha_{t|s}\mathbf{z}_t + (1 - \alpha_{t|s})\frac{\mathbf{u}}{N}\right] \odot \left[\beta_s\left(\alpha_s\mathbf{x}_\theta + (1 - \alpha_s)\frac{\mathbf{u}}{N}\right) + (1 - \beta_s)\mathbf{m}\right]}{\beta_t\left[\alpha_t\mathbf{z}_t^\top\mathbf{x}_\theta + (1 - \alpha_t)\frac{1}{N}\right]}\right) \\
& = \mathrm{Cat}\left(\mathbf{z}_s; \frac{N\alpha_t\mathbf{z}_t \odot \mathbf{x}_\theta + (\alpha_{t|s} - \alpha_t)\mathbf{z}_t + (\alpha_s - \alpha_t)\mathbf{x}_\theta + \frac{(\alpha_s - \alpha_t)(1 - \alpha_s)}{N\alpha_s}\mathbf{u}}{N\alpha_t\mathbf{z}_t^\top\mathbf{x}_\theta + (1 - \alpha_t)}\right)
\end{aligned}
\tag{16}
$$

## B.4 DIFFUSION LOSS

Consider the case $\mathbf{z}_t = \mathbf{m}$. Let us simplify $\mathrm{D}_{\mathrm{KL}}\left(q(\mathbf{z}_s \mid \mathbf{z}_t = \mathbf{m}, \mathbf{x}) \,\|\, p_\theta(\mathbf{z}_s \mid \mathbf{z}_t = \mathbf{m})\right)$:

$$
\begin{aligned}
& \mathrm{D}_{\mathrm{KL}}\left(q(\mathbf{z}_s \mid \mathbf{z}_t = \mathbf{m}, \mathbf{x}) \,\|\, p_\theta(\mathbf{z}_s \mid \mathbf{z}_t = \mathbf{m})\right) \\
& = \sum_{\mathbf{z}_s} q(\mathbf{z}_s \mid \mathbf{z}_t = \mathbf{m}, \mathbf{x}) \log \frac{q(\mathbf{z}_s \mid \mathbf{z}_t = \mathbf{m}, \mathbf{x})}{p_\theta(\mathbf{z}_s \mid \mathbf{z}_t = \mathbf{m})} \\
& = \frac{(\beta_t - \beta_s)}{N(1 - \beta_t)}\left[(N\alpha_s + 1 - \alpha_s)\log\frac{N\alpha_s\mathbf{x}_\theta^i + 1 - \alpha_s}{N\alpha_s + 1 - \alpha_s} + \sum_{j=1}^{N-1}(1 - \alpha_s)\log\frac{N\alpha_s\mathbf{x}_\theta^j + 1 - \alpha_s}{1 - \alpha_s}\right]
\end{aligned}
\tag{17}
$$

Consider the case $\mathbf{z}_t \neq \mathbf{m}$. Let us simplify $\mathrm{D}_{\mathrm{KL}}\left(q(\mathbf{z}_s \mid \mathbf{z}_t \neq \mathbf{m}, \mathbf{x}) \,\|\, p_\theta(\mathbf{z}_s \mid \mathbf{z}_t \neq \mathbf{m})\right)$:

$$
\begin{aligned}
& \mathrm{D}_{\mathrm{KL}}\left(q(\mathbf{z}_s | \mathbf{z}_t \neq \mathbf{m}, \mathbf{x}) \,\|\, p_\theta(\mathbf{z}_s | \mathbf{z}_t \neq \mathbf{m})\right) \\
& = \frac{\alpha_t'}{N\alpha_t}\left[\frac{N}{N\alpha_t\mathbf{x}^i + 1 - \alpha_t} - \frac{N}{N\alpha_t\mathbf{x}_\theta^i + 1 - \alpha_t}\right. \\
& \left. - \sum_{j=1}^{N-1}\left(\frac{N\alpha_t\mathbf{x}^j + 1 - \alpha_t}{N\alpha_t\mathbf{x}^i + 1 - \alpha_t}\right)\log\left[\frac{N\alpha_t\mathbf{x}_\theta^i + 1 - \alpha_t}{N\alpha_t\mathbf{x}_\theta^j + 1 - \alpha_t} \cdot \frac{N\alpha_t\mathbf{x}^j + 1 - \alpha_t}{N\alpha_t\mathbf{x}^i + 1 - \alpha_t}\right]\right].
\end{aligned}
\tag{18}
$$

## C  NOISE SCHEDULE

### C.1  NOISE SCHEDULE PARAMETERIZATION

Following prior works (Lou et al., 2024; Sahoo et al., 2024), we parameterize $\alpha_t = e^{-\sigma(t)}$, where $\sigma(t) : [0, 1] \to \mathbb{R}^+$. Various functional forms of $\sigma(t)$ are listed below:

**Log Linear.**  The log linear schedule is given as:
$$\sigma(t) = -\log(1 - t). \tag{19}$$

**Cosine Squared.**  The cosine squared schedule is given as:
$$\sigma(t) = -\log \cos^2\left(\tfrac{\pi}{2}(1 - t)\right). \tag{20}$$

**Geometric.**  The geometric schedule is given as:
$$\sigma(t) = (\sigma_{\min})^{1-t}(\sigma_{\max})^t, \tag{21}$$
where $\sigma_{\min}$, and $\sigma_{\max}$ are hyperparameters. In our experiments we set $\sigma_{\min} = 1 * 10^{-3}$, and $\sigma_{\max} = 1$.

### C.2  ELBO INVARIANCE TO ABSORBING NOISE SCHEDULE

The function $\beta_t$ is invertible due to the monotonicity assumption, and so we can perform the following change of variables: $\gamma \equiv \log(1 - \beta_t)$. Let $f : [0, 1] \to \mathbb{R}^-$ be a function such that $\gamma = f(t)$. Note that $\beta_t$ goes through a monotonic transformation to obtain $\gamma$; hence, $\gamma$ is also monotonic in $t$ since $\alpha_t$ is monotonic in $t$. This implies that the function $f$ is invertible. Let $t = f^{-1}(\gamma)$ a nd $\hat{\mathbf{x}}_\theta = \mathbf{x}_\theta(\mathbf{z}_{f^{-1}(\gamma)}, f^{-1}(\gamma))$. Then, when $\mathbf{z}_t = \mathbf{m}$, we can have the following $\mathcal{L}_{\text{diffu}}^m$:

$$
\begin{aligned}
\mathcal{L}_{\text{diffu}}^m &= \mathbb{E}_q \int_{t=0}^{t=1} \frac{\beta_t'}{1 - \beta_t} \left( \mathcal{F}(\mathbf{1}, \alpha_s)_i \log \frac{\mathcal{F}(\mathbf{x}_\theta, \alpha_s)_i}{\mathcal{F}(\mathbf{1}, \alpha_s)_i} + \sum_{j \neq i}(1 - \alpha_s) \log \frac{\mathcal{F}(\mathbf{x}_\theta, \alpha_s)_j}{1 - \alpha_s} \right) dt \\
&= -\mathbb{E}_q \int_{t=0}^{t=1} \left( \mathcal{F}(\mathbf{1}, \alpha_s)_i \log \frac{\mathcal{F}(\mathbf{x}_\theta, \alpha_s)_i}{\mathcal{F}(\mathbf{1}, \alpha_s)_i} + \sum_{j \neq i}(1 - \alpha_s) \log \frac{\mathcal{F}(\mathbf{x}_\theta, \alpha_s)_j}{1 - \alpha_s} \right) \frac{d\log(1 - \beta_t)}{dt} dt \\
&= -\mathbb{E}_q \int_{t=0}^{t=1} \left( \mathcal{F}(\mathbf{1}, \alpha_s)_i \log \frac{\mathcal{F}(\mathbf{x}_\theta, \alpha_s)_i}{\mathcal{F}(\mathbf{1}, \alpha_s)_i} + \sum_{j \neq i}(1 - \alpha_s) \log \frac{\mathcal{F}(\mathbf{x}_\theta, \alpha_s)_j}{1 - \alpha_s} \right) \frac{df(t)}{dt} dt \\
&= -\mathbb{E}_q \int_{\gamma=-\infty}^{\gamma=0} \left( \mathcal{F}(\mathbf{1}, \alpha_s)_i \log \frac{\mathcal{F}(\hat{\mathbf{x}}_\theta, \alpha_s)_i}{\mathcal{F}(\mathbf{1}, \alpha_s)_i} + \sum_{j \neq i}(1 - \alpha_s) \log \frac{\mathcal{F}(\hat{\mathbf{x}}_\theta, \alpha_s)_j}{1 - \alpha_s} \right) d\gamma
\end{aligned}
\tag{22}
$$

This new formulation demonstrates that $\mathcal{L}_{\text{diffu}}^m$ is invariant to the functional form of $\beta_t$. Moreover, when $\mathbf{z}_t \neq \mathbf{m}$, $\mathcal{L}_{\text{diffu}}^u$ is mathematically independent of $\beta_t$, and therefore the overall diffusion loss $\mathcal{L}_{\text{diffu}}$ is invariant to the functional form of $\beta_t$.

## D  EXPERIMENTAL DETAILS

We conduct all experiments using $8\times$ *NVIDIA A800 80G* GPUs.

### D.1  LANGUAGE GENERATION

**Text8.**  We follow standard practices (Austin et al., 2021; Lou et al., 2024) for conducting experiments on the Text8 dataset, which has a vocabulary size of 28, consisting of 26 lowercase letters, a whitespace token, and a mask token. We adhere to the standard dataset split and train MAUD using the same transformer architecture as MDLM Austin et al. (2021). Specifically, we employ a transformer architecture with 12 layers, a hidden dimension of 768, 12 attention heads, and a timestep embedding of 128. All models are trained on text chunks of length 256 for 1M steps with a batch size of 512.

**OpenWebText.** We follow the standard dataset splits (Lou et al., 2024; Sahoo et al., 2024), reserving the last 100K documents as the validation set. OpenWebText is tokenized using the GPT-2 tokenizer, resulting in a vocabulary size of approximately 50K. All models are trained on sequences wrapped to a length of 1,024, with the end-of-sequence (EOS) token set as both the first and last token in every batch. Architectural configurations remain consistent with those in the Text8 experiments: we use transformers with 12 layers, a hidden dimension of 768, 12 attention heads, and a timestep embedding of 128 where applicable. Word embeddings are not tied between the input and output. Other training configurations also remain unchanged Sahoo et al. (2024): we use the AdamW optimizer with a batch size of 512, a learning rate of 0.0003, and a linear warm-up of 2,500 steps. All models are trained for 1M steps.

### D.2 LANGUAGE UNDERSTANDING

**GSM8K.** Following the experimental setup of Gong et al. (2024), we fine-tune MAUD, which was pretrained on OpenWebText, on GSM8K. For fine-tuning, we follow the configuration of Ye et al. (2024). Specifically, we use the AdamW optimizer with a learning rate of 3e-4, and train on GSM8K with a batch size of 512 for 120K iterations.

## E ETHICS STATEMENT

This work adheres to the ICLR Code of Ethics. No human subjects or animal experimentation were involved. All datasets used in this study were obtained in compliance with the relevant usage guidelines, ensuring that no privacy violations occurred. We have taken care to minimize potential biases or discriminatory outcomes throughout the research process. No personally identifiable information was used, and no experiments were conducted that could raise privacy or security concerns. We are committed to maintaining transparency and integrity in all aspects of this work.

## F REPRODUCIBILITY STATEMENT

We have made every effort to ensure the reproducibility of our results. The experimental setup, including training steps, model configurations, and hardware details, is described in detail in the main paper and appendix. We also provide a full description of MAUD to facilitate replication of our experiments.

In addition, all datasets used in this work are publicly available, ensuring consistent and reproducible evaluation results. We believe these measures will enable other researchers to reproduce our findings and further advance the field.

## G USE OF LARGE LANGUAGE MODELS

In this work, large language models were used solely for grammar correction and writing refinement. Specifically, we employed ChatGPT-5[2], with the following prompt: *"Please act as a professional writer to correct the grammatical errors in the input text and polish the writing. Please note that you should not alter the original meaning of the text in any way; your task is only to refine the writing style to make it more professional and academic."*

Importantly, the LLM was not involved in the ideation, research methodology, or experimental design. All concepts, methods, and analyses were conceived and executed by the authors. The LLM's contributions were limited to improving the linguistic presentation of the manuscript, without any involvement in the scientific content or data analysis.

The authors take full responsibility for the content of this manuscript, including any text that was generated or refined with LLM assistance. We have ensured that the use of LLMs adheres to ethical guidelines and does not contribute to plagiarism or scientific misconduct.

---

[2]https://chatgpt.com/

