# OpenReview forum: "Keep Refining Your Discrete Diffusion Model: A Mixture of Absorbing and Uniform Processes"
_ICLR.cc/2026/Conference — Submitted to ICLR 2026_

### Official Review · Reviewer_v9uv · 2025-10-15

**Soundness:** 2
**Presentation:** 2
**Contribution:** 1
**Rating:** 2
**Confidence:** 5

**Summary:**

This paper introduces the Mixture of Absorbing and Uniform Diffusion (MAUD) model. The authors are motivated by the limitations of two dominant discrete diffusion frameworks: Uniform Discrete Diffusion, which alters tokens too abruptly, and Masked Discrete Diffusion, which cannot further modify a token once it has been masked. To address these issues, MAUD interpolates between the two paradigms by designing separate noise schedules for each and allowing both noising processes to occur simultaneously. The authors evaluate MAUD across a range of datasets, including Text8, LM1B, OpenWebText, and various language understanding benchmarks.

**Strengths:**

The paper is well written and self-contained, with a clear overall structure. The motivation is well articulated and effectively demonstrated. The methodology is clearly presented, with all theoretical aspects carefully developed. The authors also conducted extensive experiments to validate the performance of their proposed model.

**Weaknesses:**

The authors claim, as their first contribution (line 56), to have identified the limitations of pure uniform and absorbing forward processes. However, the inability to unmask tokens is a well-known shortcoming of traditional absorbing diffusion, and prior works—such as UDLM—have already explicitly discussed this issue. Therefore, it is unconvincing for the authors to take credit for “identifying” this problem.

Furthermore, interpolating between uniform and absorbing diffusion is not a novel idea. As early as D3PM, one of the pioneering works on discrete diffusion models, the concept of mixing uniform and absorbing processes was already explored (see Appendix A.2.6). While the MAUD formulation may introduce more sophisticated technical details, the underlying spirit remains essentially the same.

**Questions:**

The major concerns are summarized in the Weaknesses section. The most significant issue lies in the lack of clarity regarding the differences between the proposed formulation and D3PM. The authors should provide a more thorough explanation and analysis to convincingly demonstrate how their method meaningfully departs from or improves upon D3PM.

---

### Official Review · Reviewer_sJoZ · 2025-10-16

**Soundness:** 2
**Presentation:** 4
**Contribution:** 1
**Rating:** 2
**Confidence:** 5

**Summary:**

This paper investigates combining absorbing and uniform diffusion processes for discrete diffusion models. Starting with a review of the most popular types of discrete diffusion, absorbing diffusion (a.k.a. masked diffusion) and uniform diffusion, the paper highlights some of the unique challenges associated with either purely absorbing or purely uniform diffusion. The paper then goes on to propose a solution to these problems by combining the two types of noise. Specifically, an interpolation between absorbing and uniform diffusion is proposed, with the aim of generating sequences by first unmasking missing tokens and later transitioning to refinement through token replacement. The forward and reverse process, and the diffusion ELBO is also derived, providing the necessary tools for training and inference.
The proposed method, called MAUD, is evaluated on a variety of language modeling tasks, ranging from density estimation on Text8 and OpenWebtext (OWT), to zero-shot perplexity (PPL) estimation on held-out data, to natural language understanding in the form of multiple-choice questions and simple math problems. Among the evaluated methods, MAUD performs competitively or slightly surpasses mask-only diffusion baselines.

**Strengths:**

### S1. Writing
The paper is well-written and easy to follow, with the provided illustrations (Fig. 2 and 3) concisely explaining the proposed approach in an intuitive and easy-to-grasp format.
The limitations of existing approaches are laid out clearly, motivating the proposed approach as an obvious and sensible step.

### S2. Sound derivation
The derived forward and reverse distributions, as well as the ELBO for the proposed diffusion process appear to be sound and correct.

### S3. Novel variant of mask-and-replace diffusion
Compared to prior work (Gu et al., 2022; von Rütte et al., 2025), the proposed mask-and-replace diffusion process mixes absorbing and uniform diffusion at different rates. While Gu et al. (2022) have a constant proportion of uniform noise and von Rütte et al. (2025) have an increasing-decreasing uniform noise schedule, the proposed approach ablates three new schedules of adding uniform noise.

**Weaknesses:**

(Ordered by decreasing severity)

### W1. Lacking novelty and missing comparisons to prior work
The presented paper bears a striking similarity to GIDD (von Rütte et al., 2025), in that both papers identify the same limitations of purely absorbing diffusion and propose the same solution, which is to combine absorbing and uniform noise to obtain a mask-and-replace diffusion process. Nevertheless, a comparison between MAUD and GIDD, qualitatively or quantitatively, is missing. To my understanding, GIDD introduces a general theory for interpolating diffusion processes, potentially making the proposed process a special case of GIDD.

Furthermore, the authors cite Gu et al. (2022) in the context of limitations of UDM, when, in fact, one of Gu at al.’s key contributions was a mask-and-replace diffusion process that is essentially the same as the one proposed in this work.

This undermines core contribution claims of this paper: The limitations of purely absorbing or uniform diffusion (L56) have already been identified by prior work (Gu et al., 2022; von Rütte et al., 2025) and the proposed interpolating diffusion process (L57) is not novel (Gu et al., 2022; von Rütte et al., 2025).

In similar fashion, while MDLM (Sahoo et al., 2024) is cited extensively for masked diffusion, concurrent work by Shi et al. (2024), MD4, is only cited in passing and not considered as a baseline despite achieving better results on Text8 and OWT.

### W2. State-of-the-art claims
The state-of-the-art claims (L60-61; L413; L422; L428) are inaccurate in multiple aspects. The reported BPC of 1.36 on Text8 is subpar to that of MD4 (Shi et al., 2024), which is reported to be 1.34. For the results on zero-shot PPL (Table 2), there exist larger diffusion language models, trained on more data, that outperform the reported results (Nie et al., 2024; Nie et al., 2025). Therefore it is crucial to consider _compute-matched_ performance. Regarding compute-matched models, performance comparison between MAUD and GIDD is only possible on OWT val. PPL (Table 4) and HSwag, WinoG, and PIQA accuracy (Table 3). Of these, GIDD outperforms MAUD on OWT PPL and WinoG while being trained on at most 232B tokens (compared to MAUD’s 524B tokens), making the performance similar at best. Furthermore, the comparison on language understanding (Table 3) does not include a strong parameter- and token-matched autoregressive baseline (e.g. Llama 127M), which further undermines the “state-of-the-art [...] among models with similar parameter budgets” (L429).

In conclusion, all of the state-of-the-art claims are partially inaccurate, requiring additional caveats (e.g. compute-matched setting, among diffusion models) and/or performance comparisons.

### W3. Sample quality evaluation
Unconditional sample quality, while difficult to measure, is an important aspect of generative models and has been extensively reported in the discrete diffusion literature. For the sake of comparability, sample quality in terms of generative PPL and sequence entropy should be reported.

### W4. Noise schedule ablation
As far as I can tell, the ablated noise schedules essentially regulate the amount of uniform noise at any given time during the diffusion process, with the geometric schedule injecting the least amount of uniform noise. This poses an obvious followup question that is left unanswered by the paper: Namely, what is the importance of this variable? E.g., what effect does varying $\sigma_\mathrm{max}$ have? What is the total amount of signal, masking, and uniform noise over time for the respective noise schedules? And how does the proportion of masking/uniform noise change over time?

### W5. Likelihood bounds
The proposed likelihood bounds appear unnecessarily complex and cumbersome, and the derivation is sparse in detail with some crucial steps missing. For example, in App. B, computing Eq. 17 by plugging in Eq. 15 involves multiple logical steps that are hard to follow as a reader without rederiving the expression manually on paper. Same goes for Eq. 18. Furthermore, Eq. 18 involves the derivative $\alpha_t’$, seemingly coming out of nowhere. Commonly, the derivative enters the picture when taking the continuous-time limit. This step is entirely missing from the derivation.

More broadly speaking, the resulting likelihood bounds are unwieldy in a way that seems preventable. For example, there exist closed-form continuous-time ELBO solutions for a wide range of diffusion processes (Campbell et al., 2022; Lou et al., 2023; von Rütte et al., 2025), which raises the question why the choice was made not to use existing theory? Using existing theory could help streamline the derivation and simplify teh resulting likelihood bounds.

---

### References
1. Austin et al., 2021: https://arxiv.org/abs/2107.03006
2. Campbell et al., 2022: https://arxiv.org/abs/2205.14987
3. Gu et al., 2022: https://arxiv.org/abs/2111.14822
4. Lou et al., 2023: https://arxiv.org/abs/2310.16834
5. Nie et al., 2024: https://arxiv.org/abs/2410.18514
6. Nie et al., 2025: https://arxiv.org/abs/2502.09992
7. Sahoo et al., 2024: https://arxiv.org/abs/2406.07524
8. Shi et al., 2024: https://arxiv.org/abs/2406.04329
9. von Rütte et al., 2025: https://arxiv.org/abs/2503.04482

**Questions:**

- Q1. What are the similarities and differences between the proposed method and GIDD (von Rütte et al., 2025)? Are there any key advantages to the proposed method over GIDD? (also see W1)
- Q2. Given the above remarks, which state-of-the-art claims do you maintain? Please provide additional evidence and apples-to-apples comparisons to the relevant baselines for any maintained claims. (also see W2)
- Q3. What is the sample quality of the proposed model, and how does it vary in the number of denoising steps? (also see W3)
- Q4. What is the effect of the proportion of uniform noise on the model performance? (also see W4)
- Q5: Can the proposed diffusion process be framed as a instance of GIDD? If so, would this simplify the likelihood bounds and their derivation? (also see W1, W5)

Nits (not considered in the final score):
1. L37: "greater controllability" is rather vague and warrants a citation and/or clarification.
2. L37-38: "distinct advantages for structured sequence modeling and complex logical reasoning" is also very vague. What are these distinct advantages? What is meant by "structured" sequence modeling (the cited papers are primarily concerned with natural language and DNA sequences)? What is meant by "complex" logical reasoning, as opposed to regular logical reasoning?
3. The literature has largely converged on referring to "absorbing diffusion models" as "masked diffusion models", or MDMs. For the sake of convention, it may be worth adopting that terminology. (“Masking And Uniform Diffusion”)
4. L83: Missing citation(s) for "commonly used" noise schedules.
5. L104: The uniform diffusion process was originally proposed by Austin et al. (2021), which should be cited in addition to Schiff et al. (2024).
6. L130: The absorbing diffusion process was also originally proposed by Austin et al. (2021), which should be cited.
7. L156-157: The cited reference (Gu at al., 2022) does not, as far as I can tell, discuss the increased difficulty of uniform diffusion compared to masked diffusion.
8. Section 3.2: Equations are unnumbered for no apparent reason. Generally, it’s considered good practice to number all equations for the sake of referenceability.
9. Throughout: Redefining $\mathbf{u}$ as $\mathbf{u} = \frac{\mathbf{1} - \mathbf{m}}{N}$ may simplify a lot of the equations and derivations while also being more interpretable ($\mathbf{u}$ is the categorical uniform distribution over the vocabulary).
10. L291: Algorithm 1 is called “UMDM” sampling, it seems like this should be “MAUD” sampling.

---

### Official Review · Reviewer_Loog · 2025-10-27

**Soundness:** 2
**Presentation:** 3
**Contribution:** 2
**Rating:** 4
**Confidence:** 5

**Summary:**

This paper introduces the Mixture of Absorbing and Uniform Diffusion (MAUD) model, an approach for discrete diffusion models (DDMs) that addresses the trade-off between semantic stability and token-level refinement. Purely Absorbing Diffusion Models (ADMs) (e.g., MDLM) ensure sequence-level semantic stability but cannot correct tokens once unmasked. Conversely, Uniform Diffusion Models (UDMs) allow token-level refinement but risk abrupt meaning changes at the sequence level. MAUD constructs a state transition matrix that interpolates between these two processes using two noise schedules, $\alpha_t$ (uniform) and $\beta_t$ (absorbing). This hybrid approach is shown to achieve promising performance in language modelling.

**Strengths:**

- The paper is well-written and easy to follow. The idea of applying a mixture of masking and uniform processes is intuitive, effectively combining the strengths of both approaches in discrete diffusion models.
- Experimental results demonstrate promising performance compared to other discrete diffusion models.

**Weaknesses:**

My primary concern is the connection to GIDD [1]. The proposed UMDM formulation appears to be essentially the same as GIDD (see Equation 17 in [1]). However, the paper neither cites GIDD nor discusses the relationship between the two methods.

Addressing this connection is important for clarifying the novelty of UMDM. It would be helpful for the authors to explicitly compare UMDM and GIDD, highlighting any differences in theory or implementation, and to discuss why UMDM offers advantages over or complements the existing GIDD approach.

[1] von Rütte, Dimitri, et al. "Generalized interpolating discrete diffusion." *arXiv preprint arXiv:2503.04482* (2025).

**Questions:**

The paper is very well-written, and I don’t have many questions. My only curiosity is how the $\beta$ schedule affects performance, as $\beta$ appears to control the strength of the mask and the amount of uniform noise.

---

### Official Review · Reviewer_TrWw · 2025-10-31

**Soundness:** 3
**Presentation:** 3
**Contribution:** 3
**Rating:** 6
**Confidence:** 3

**Summary:**

This paper proposes MAUD, a discrete diffusion model that linearly interpolates between the uniform diffusion models and absorbing diffusion models. The key idea is to design a transition kernel that allocates some probability mass to an absorbing [MASK] state while still permitting uniform token transitions for gradual token-level refinement. The authors derive the forward and reverse processes, give continuous-time ELBO limits, and provide the training and sampling algorithm. Empirically, MAUD improves Text8 bits-per-character, reduces perplexity across several language corpora, and shows competitive or better performance on a suite of understanding/Reasoning tasks.

**Strengths:**

1. Clear motivation. The paper analyzes complementary failure modes of UDMs and ADMs, motivating a unifying process.
2. Theoretical soundness. The mixture transition matrix and the resulting reverse-process cases are derived cleanly. The paper analyzes the ELBO in the continuous-time limit, separates masked/unmasked diffusion losses, and shows that the diffusion loss is invariant to the absorbing noise schedule β_t.
3. Better performance. MAUD improves Text8 BPC over prior DDMs and narrows the gap to AR models.
4. The paper is well-written and easy to follow.

**Weaknesses:**

1. Novelty vs. general D3PM framework. I like your paper writing and framework. However, it must be said that D3PM already permits arbitrary categorical transition kernels; MAUD’s kernel is an interpretable special case that interpolates between two priors. The paper positions this as a new model, but the conceptual step could be seen as a well-motivated instantiation inside known frameworks rather than a fundamentally new paradigm. Clarifying where MAUD lies relative to prior ratio-estimation UDMs/ADMs and guidance mixtures would help.
2. Model Scale. The model scale is limited. It’s unclear how MAUD behaves against larger recent dLLMs under matched training budgets.
3. Lack of some ablations. sensitivity to the geometric schedule’s σ_min/σ_max and to the relative pacing between α_t and β_t is not fully explored
4. Minor Typo. Algorithm 1 is titled “UMDM Sampling,” which looks like a typo.

**Questions:**

1. How do PPL/BPC and accuracy change with the number of denoising steps? Is MAUD more/less sample-efficient than a pure ADM or UDM at the same compute? (Please include wall-clock.)
2. With larger models/datasets, do MAUD’s gains persist or widen? Any observations on long-sequence generation?

---

### Meta-Review · Area_Chair_2nk6 · 2025-12-31

**Summary:**

The most serious reviewer concern, raised by both reviewers Loog and sJoZ, is the similarity to [GIDD](https://proceedings.mlr.press/v267/von-rutte25a.html) published in ICML 2025, which was not covered in the submission. There were also other concerns about incremental novelty to D3PM, and over-claiming.

**Reviewer Concerns:**

There were no author comments to address reviewer concerns, so all concerns remain outstanding.

**Reviewer Scores:**

The original reviewer scores were 6, 4, 2, 2. Given the lack of author responses, and shared awareness of the missing comparison to GIDD, it is very likely that reviewer scores will go down and unanimously lean to reject.

---

### Decision · Program_Chairs · 2026-01-26

Reject